# The Emerging Role of Tractography in Deep Brain Stimulation: Basic Principles and Current Applications

**DOI:** 10.3390/brainsci8020023

**Published:** 2018-01-29

**Authors:** Nelson B. Rodrigues, Karim Mithani, Ying Meng, Nir Lipsman, Clement Hamani

**Affiliations:** 1Sunnybrook Research Institute, Toronto, ON M4N 3M5, Canada; nelson.rodrigues@mail.utoronto.ca (N.B.R.); karim.mithani@mail.utoronto.ca (K.M.); ying.meng@medportal.ca (Y.M.); Nir.Lipsman@sunnybrook.ca (N.L.); 2Division of Neurosurgery, Sunnybrook Health Sciences Centre, University of Toronto, Toronto, ON M4N 3M5, Canada; 3Institute of Medical Sciences, University of Toronto, Toronto, ON M5S 1A8, Canada; 4Harquail Centre for Neuromodulation, Sunnybrook Health Sciences Centre, Toronto, ON M4N 3M5, Canada

**Keywords:** diffusion tensor imaging, tractography, deep brain stimulation, tremor, depression

## Abstract

Diffusion tensor imaging (DTI) is an MRI-based technique that delineates white matter tracts in the brain by tracking the diffusion of water in neural tissue. This methodology, known as “tractography”, has been extensively applied in clinical neuroscience to explore nervous system architecture and diseases. More recently, tractography has been used to assist with neurosurgical targeting in functional neurosurgery. This review provides an overview of DTI principles, and discusses current applications of tractography for improving and helping develop novel deep brain stimulation (DBS) targets.

## 1. Introduction

Deep brain stimulation (DBS) has been successfully used to treat different neuropsychiatric disorders. As one of its mechanisms is to modulate circuit activity and fibre pathways in the vicinity of the electrodes [1,2], the use of tractography during targeting has been advocated in some applications of this technique. Despite the more frequent use of tractography, many neurosurgeons, neurologists and psychiatrists still do not have a good understanding of its potential use during DBS targeting.

In this article, we provide a basic review of the principles behind diffusion tensor imaging and tractography and describe some of the most common clinical applications of tractography-based DBS targeting, namely tremor and depression.

## 2. Tractography

Diffusion tensor imaging (DTI) is an MRI-based technique that leverages the natural diffusion of water molecules to map white-matter tracts in the brain [3]. Visualization and segmentation of these tracts have implications in various aspects of clinical neuroscience, one of them being neurosurgical targeting. 

Diffusion is the random motion of water molecules, which is a result of their rapid movement and frequent collisions with surrounding molecules and structures [4]. This random motion in suspended fluid with no chemical or mechanical gradient is known as Brownian motion. Assuming no adjacent barriers exist, diffusion movements of water molecules along any axis have a Gaussian distribution.

In diffusion-weighted magnetic resonance imaging (dwMRI), the applied magnetic field is non-uniform in either the *x*-, *y*-, or *z*-axes. This single-axis heterogenic “dephasing” field results in varying motion of water molecules, depending on the strength of the field. After a brief time interval (typically 10–100 ms), a non-uniform “rephasing” magnetic field is applied in the exact opposite pattern. If water molecules have identical positions in the axis of the applied gradient at the time of rephasing and dephasing, the ultimate MRI signal should be unvaried from start to finish. However, if water molecules change position between these two time-points, the signal will be quantifiably attenuated.

dwMRI is typically acquired with three orthogonal three-dimensional gradients, one in each of the *x*-, *y*-, and *z*-axes. This allows for estimating the apparent diffusion coefficient (ADC) of water in the brain, which quantifies the flux of these molecules under equilibrium conditions [5,6]. This model does not, however, account for variations in the amount of diffusion of water in different directions—a property known as diffusion anisotropy. Anisotropy is a result of tissue microstructure, either normal or pathologic, limiting the motion of water in one or more directions. In particular, highly organized tissue architecture, as well as certain types of tissue (e.g., axon membranes and microtubules) result in water undergoing anisotropic diffusion.

Intuitively, this signal attenuation will be maximal when the applied gradient is aligned in the same orientation as the primary direction of water diffusion. For example, fibres in the corpus callosum—oriented mediolaterally—and those in the superior longitudinal fasciculus—oriented anteroposterioriorly—will exhibit maximal signal attenuation at approximately perpendicular gradients. In fibres oriented obliquely to the direction of the applied gradient, the amount of signal attenuation will be proportional to the amount of water diffusion that is parallel to the gradient.

Measuring ADCs of water along at least six different gradients allows for mathematically modelling its three-dimensional molecular diffusivity at any voxel in the acquired image. This model is a symmetric 3 × 3 matrix composed of six distinct parameters, and is known as the diffusion tensor. Through an operation known as eigendecomposition, the diffusion tensor can be rendered as eigenvectors (e1, e2, e3) representing the direction of diffusion, and eigenvalues (λ1, λ2, λ3) representing the magnitude of diffusion in each of those directions (Figure 1). Each eigenvector corresponds to an eigenvalue (e.g., e1 corresponds to λ1, etc.), and a strict ordering of the eigenvalues (λ1 ≥ λ2 ≥ λ3) is maintained. Accordingly, e1—corresponding to the largest eigenvalue—is the dominant orientation of diffusion of water in an anisotropic voxel, and is known as the “principal eigenvector”. This representation of the diffusion tensor can be visualized as a scalene ellipsoid, with the long axis being the dominant direction of water diffusion (i.e., the principal eigenvector), and the short axis being the least traversed vector. Since the diffusion of water in the brain is predictably limited by a small number of structures, the combination of diffusion tensors and their principal eigenvectors across many voxels provides an estimation of neuronal white matter architecture. Notably, this conclusion assumes that there is only a single, unidirectional, homogeneous bundle of axons in each voxel, which can be problematic as will be discussed later. The precision of the diffusion tensor model is improved by acquiring diffusion-weighted images in more directions. However, this also increases the number of acquisition and processing iterations required.

One of the most useful features of DTI is its quantitative nature. Using only a few summary values, the white matter architecture of individuals or groups of individuals can be compared in selected brain regions (e.g., before and after a surgical procedure). The most commonly used values in clinical studies are the mean diffusivity (MD), and fractional anisotropy (FA). MD is the mean of the eigenvalues, and represents the average size of the diffusion tensor, independent of anisotropy or orientation. FA summarizes the degree of anisotropy in a particular voxel, and is a measure of the tensor shape, independent of size or orientation. Changes in these values may suggest neuronal pathology, such as cell death. Importantly, these are simplified, summary metrics, and assume many caveats that are not reflective of real changes in tissue micro-architecture; they should therefore be interpreted cautiously. In other words, quantitative DTI metrics should only be used as a potential indicator of changes in tissue microstructure, and not as a direct measure of it.

The spatial resolution of diffusion-weighted images is typically 2 mm, meaning that a typical voxel contains ~10^5^ axons [7]. DTI-based tractograpy is therefore not intended to map individual axons, but rather to provide an estimation of the trajectory of white matter tracts. A significant limitation of DTI fibre tracking is the inability to accurately delineate heterogeneous tracts within a single voxel. Kissing, crossing, converging, or diverging tracts have differing fibre orientations, each of which contributes to the voxel’s computed diffusion tensor [8,9]. The unrealistic assumption of a single homogeneous, unidirectional fibre population per voxel employed in DTI can considerably limit the accuracy of fibre tracking, considering that up to one third of white-matter voxels in the brain are estimated to contain multiple fibre orientations [10]. Furthermore, the relatively low spatial resolution of dwMRI can further impair the geometric accuracy of fibre tracts and the ability to distinguish between multiple, closely associated bundles. Newer, multi-fibre approaches have demonstrated greater sensitivity in fibre tractography using dwMRI, and include: high angular resolution diffusion imaging (HARDI) [11], Q-Ball imaging (QBI) [12], hybrid diffusion imaging (HYDI) [13], diffusion spectrum imaging (DSI) [14], spherical deconvolutional models [15], composite hindered and restricted model of diffusion (CHARMED) [16], persistent angular structure (PAS) [17], and diffusion orientation transformation (DOT) [18]. Most of these approaches are more demanding on computational resources and acquisition requirements, and many are still being refined for clinical uptake [19]. Accordingly, DTI tractography remains a widely used modality for surgical planning, and validation studies have found a reasonable degree of correspondence between DTI-based white matter tracts and histologically identified axon bundles [20,21].

Broadly, two main approaches to tractography are commonly employed in clinical and research settings: deterministic and probabilistic.

Deterministic tractography computes white matter tracts based on: (1) user-specified seed point(s), (2) the principal eigenvector of diffusion, as determined by the tensor at each voxel, and (3) user-specified thresholds that restrict the tract’s trajectory. Seed points are particular voxels of interest selected by the user through which the desired tract is believed to pass. Users can specify one or more seeds that are known to be anatomically connected to the tract of interest (i.e., a “start point”, and often “mid points” and “end points”). A bidirectional streamline will then be formed, starting at the first seed point, following the local principal eigenvector field to the next adjacent voxels. In an interactive manner, the local principal eigenvectors of diffusion are used to elongate the streamline, up until a point where the diffusion tensor in the voxel is considered too different from the tract to be a part of it. This terminal point is based on user-specified, heuristically determined thresholds, most typically the curvature angle of the tract and the local FA. A sharp curve or bend in the principal eigenvector likely suggests the presence of a different white matter tract. The exact curvature threshold is pre-determined (e.g., between 30°–70°). In general, a “low” FA suggests the termination of a white matter tract; the threshold is once again pre-determined (e.g., >0.2). A commonly used strategy is to initially set relatively restrictive thresholds and gradually ease them as necessary, using real-time feedback to empirically identify the optimal settings. The ultimate tractographic parameters used should, however, be logged to ensure downstream reproducibility and valid comparisons. Importantly, common clinically approved and commercially available DTI software, typically integrated into surgical navigation suites, employ deterministic approaches. The most commonly used tractography algorithm in these products is fiber assignment by continuous tracking (FACT). Examples of such software includes Medtronic’s StealthViz^®^ and StealthDTI^®^ (Medtronic, MN, USA) modules, and NordicNeuroLab’s nordicBrainEx (Bergen, Norway) [22].

Although the deterministic tractography consists of a relatively simpler approach, estimating the orientation at every voxel is rarely entirely accurate. Each voxel has a range of potential orientations, and a deterministic approach considers the mean of these to be the “true” orientation. There is a standard error associated with this mean, and these errors are propagated throughout the tract. As a result, considerable uncertainty in white matter trajectory can exist.

Rather than relying solely on the orientation of the principal eigenvector, probabilistic approaches consider the entire distribution of possible orientations in all voxels emerging from the specified seed point. The mean of this distribution is the orientation used in deterministic tracography. The width of the distribution is proportional to the associated uncertainty. Accordingly, numerous tracts—in the order of 10,000—are mapped from the seed point. The number of tracts crossing each voxel is then computed, a probability map is derived, and voxels with the most number of tracts passing through them are included in the elongating streamline.

Probabilistic tractography provides more sophisticated and robust results, but is also more computationally demanding, harder to perform, and less intuitive to interpret than its deterministic counterpart. Figure 2 shows a schematic of deterministic and probabilistic approaches, and how they differ.

Although probabilistic tractography is lauded for taking into account the uncertainty associated with diffusion imaging data; it has been criticized for its susceptibility to minor changes in data noise. Global tractography was developed as an alternative to deterministic and probabilistic algorithms, with the intention of further improving the robustness of fibre tracking using diffusion imaging [23]. This approach aims to construct the fibre configuration that best explains the dwMRI data. Building on the assumption that axons tend to run in gently bending organized fascicles, global tractography delineates the local fibre orientation of an ambiguous voxel by examining the estimated orientations in the voxel’s spatial neighbourhood. Although global tractography is more resilient to changes in acquisition noise than the aforementioned streamline methods, it is computationally demanding and requires fairly specific microstructural models and rigid parameters that are not always readily adaptable to available data [24]. Nevertheless, advancements are being made in this technique to help augment the reliability and utility of dwMRI tractography [24,25].

An important criticism of DTI fibre tracking is that the existence of numerous distinct approaches to data processing and analysis can lead to considerable variability in results [25,26]. The absence of standardization and difficulties with reproducibility reduces the clinical utility of DTI tractography. It is imperative to emphasize that DTI fibre tracking should be scrutinized and treated with reasonable skepticism, serving only as one of many layers of evidence in surgical decision-making.

## 3. Tractography and DBS

The general notion that tractography may help to increase accuracy when treating patients. Mapping tracts that may be important for the efficacy and side effects of DBS has been recently explored in a number of applications. In addition to major tracts, fibre bundles that cannot be viewed with traditional magnetic resonance imaging, such as the dentato-rubro-thalamic tract (DRT), can be identified as optimal DBS targets. In this section we will consider two conditions for which tractography has been proposed to improve clinical outcomes—tremor and treatment refractory depression (TRD).

### 3.1. Tremor and Movement Disorders

Essential tremor is the most prevalent movement disorder in adults. It presents as a bilateral postural or kinetic tremor that can severely diminish quality of life. DBS is a safe modality for the treatment of refractory essential tremor (ET) and other forms of tremors. Neurosurgeons have found success stereotactically implanting DBS leads in the ventral intermediate nucleus (Vim) of the thalamus [27,28] and adjacent regions, such as the posterior subthalamic area (PSA) [29,30,31,32].

The DRT is a compact fibre bundle that plays an important role in the coordination of somatomotor function [33]. The tract mainly originates in the dentate cerebellar nucleus, ascends through the upper pons, decussates, and travels through the red nucleus prior to innervating the ventralis oralis posterior nucleus and the Vim [33]. Bearing in mind the proximity of the DBS electrodes to the DRT in patients successfully treated for tremor [34], tractography data has led to the hypothesis that modulation of this tract could be important for the effects of stimulation (Figure 3) [33]. This possibility has been tested in open label clinical trials using tractography for targeting [33,35] and in a series of patients considered as being non-responders who benefited from DBS when electrodes were re-implanted near the DRT [36]. At present, a double-blind randomized clinical trial comparing the efficacy of traditional Vim and DRT DBS is under way [37].

In the last few years, tractography and DTI studies have also been conducted in patients with Parkinson’s disease treated or referred for DBS. One of the postulated mechanisms for the effects of subthalamic (STN) stimulation is the modulation of the hyperdirect pathway between the nucleus and cortical regions [38]. A recent study compared tractography-defined hyperdirect pathways using a tensor-based deterministic approach and an advanced probabilistic method based on constrained spherical deconvolution [39]. Though both identified connections between the ipsilateral motor cortex and the STN, a > 1.4 mm difference was found between methods when the target centre of mass was considered. While probabilistic tractography was associated with fairly continuous reconstructed connections terminating in the dorsolateral STN, these were sparser and presented variable subsets when the tensor-based method was used [39].

Connectivity patterns between cortical and STN cluster regions associated with improvement in clinical symptoms were also studied with high angular resolution diffusion and probabilistic tractography [40]. Good tremor improvement was associated with cortical connectivity between the primary motor cortex and an STN cluster corresponding to the posterior, superior and lateral portion of the nucleus. Improvement in bradykinesia corresponded to clusters near the superior STN border in a medial-posterior location with connectivity being observed with the supplementary motor area [40]. The rigidity cluster extended from the region described for bradykinesia to the STN-zona incerta border. Predictive connectivity was observed with the prefrontal cortex [40]. In a different study, clinical response correlated with connectivity between the DBS electrode and a network of brain regions, including structural connectivity to supplementary motor areas and functional anti-correlation to the primary motor cortex [41]. These results suggest that connectivity may become an important tool for DBS targeting in the future. In addition to the STN, tractography has also been used to define fibre pathways near the pedunculopontine nucleus [42,43]. To date, however, no study has used this modality to actually target regional structures.

### 3.2. Treatment Resistant Depression

The success and safety of DBS in ameliorating movement disorders has led to its investigation as a putative treatment for psychiatric conditions, including treatment of refractory depression. Initial open label clinical trials in which DBS was delivered to targets such as the subgenual cingulate gyrus (SCG), ventral capulse/ventral striatum (VC/VS) and others have shown promising results [44,45,46,47]. However, these have not been confirmed in blinded assessments comparing active vs. sham stimulation [48,49]. Several possibilities have been raised to account for these discrepancies, and multiple novel strategies have been proposed to improve clinical outcome. One of those is to refine targeting strategies. As some DBS targets are located in regions predominantly composed of white matter fibres, investigators have studied whether tractography could be employed to refine electrode placement.

A set of studies using tractography-based DBS in TRD was conducted by investigators bilaterally targeting the supero-lateral branch of the medial forebrain bundle (slMFB) (Figure 4) [50]. The rationale for the procedure was based on imaging studies conducted in Parkinson’s disease patients undergoing surgical treatment [51]. DBS delivered through electrodes implanted medially in the subthalamic region may induce dysphoria and mania [52,53]. A potential anatomical substrate involved in these responses has been attributed to inadvertent stimulation of the medial forebrain bundle (MFB) [51,54,55]. In a tractography study, the authors have shown a tract departing the midbrain that bifurcated into inferomedial and superolateral branches [54,55]. The superolateral branch spreads out laterally, undercutting the thalamus to ascend to the inferior portion of the anterior limb of the internal capsule, where it intermingles with the anterior thalamic radiation [54,55]. Guided by tractography, a trial was conducted in seven patients with TRD bilaterally implanted with electrodes in the region of the medial forebrain bundle [50]. Remarkably, six patients experienced robust and rapid clinical improvement within hours or days after stimulation onset, four of whom were classified as remitters at the last observation period (12–33 weeks post-surgery) [50]. In addition to short-term results, positive outcomes have been demonstrated at the last clinical follow-up, up to four years after surgery [56].

Riva-Posse and colleagues used whole brain activation volume probabilistic tractography to study tracts travelling through the subgenual region [57]. In that series, six out of 16 patients had at least 50% decrease in depression scores (i.e., responders) at six months, while ten were considered to be non-responders. After changing stimulation parameters, two of these six patients responded to treatment [57]. When analysing tractography data, the authors noticed that electrode contacts in responders were modulating a blueprint of fibre tracts, including the forceps minor, cingulum bundle and projections to subcortical nuclei [57] (Figure 5). In non-responders, these fibres did not seem to be modulated. In one non-responder who benefitted from a change in stimulation settings, the investigators activated a contact within the blueprint [57].

Following this initial work, Riva-Posse et al. have studied a cohort of 11 TRD patients using deterministic tractography to preplan electrode implantation [58]. The choice of stimulation contacts was based on post-operative probabilistic tractography [58]. After one year, nine of the eleven patients were considered to be responders, six of whom achieved remission [58]. This study provides further support to the notion that tractography-guided DBS may be important for a good postoperative outcome.

In an additional report aimed to replicate and extend our understanding of MFB-targeted DBS, four patients were followed for 52 weeks after tractography-based implantation [59]. Electrodes were activated after four weeks of single-blinded sham-stimulation. Despite the small sample size, the study reported that three of the four patients rapidly became euthymic within seven days of modulating the MFB [59].

Overall, the short-term and sustained antidepressant response produced by MFB stimulation in TRD is noteworthy. Although conclusions from these studies should be considered with caution due to the small sample size and lack of control groups, they exemplify how tractography can be used to generate hypotheses as to what structures may be modulated by DBS and refine targeting.

## 4. Conclusions

With the increased knowledge about the importance of fibre pathways for DBS surgery, a growing number of tractography applications have been proposed. Despite the promising work described in our review, we note that, as currently used, tractography is not without limitations. For one, tractography is unable to differentiate between afferent and efferent fibre tracts. As a result, neurosurgeons must rely on anatomical knowledge to determine the best way to modulate certain areas. Certain imaging paradigms are capable of determining directionality, such as magneto-encephalography, and may represent a multi-modal approach to understanding disease circuits [60]. Another cause for variability is noise, which can lead to uncertainty in the results. Clinicians must find a good balance between time in the scanner and practicality, as elongated scan times may cause patients to feel restless, producing motion artefacts [61]. Typically, poor image quality can be overcome by using a stronger magnetic field or with multiple DTI acquisitions [61]. Overall, white matter tractography can be time-consuming and may require specialized analysis and skillsets that may limit the accessibility of tractography-based DBS.

These limitations notwithstanding, tractography-based DBS is a feasible option over traditional DBS. Its role has been crucial in developing our understanding of the role white matter plays in disease pathology, for example, the MFB has become a target of interest in depression [50,59]. If the rapid relief in depression through MFB stimulation is confirmed and replicated in large randomized sham-controlled studies, it could potentially be a ground-breaking treatment for intractable depression. 

Leveraging tractography techniques as a tool to refine DBS may also allow clinicians and researchers to better understand the aetiology behind mood disorders. Many reviews focus on targetable brain regions that may play a role in depression mentioning the role of white matter in passing [62]. However, with the rising interest in connectomics and the Human Connectome Project, the importance of fibre tracts is receiving recognition as an important factor in depression. To this end, tractography will hopefully become a commonly employed tool by neurosurgeons to ameliorate these complex psychiatric conditions.

## Figures and Tables

**Figure 1 brainsci-08-00023-f001:**
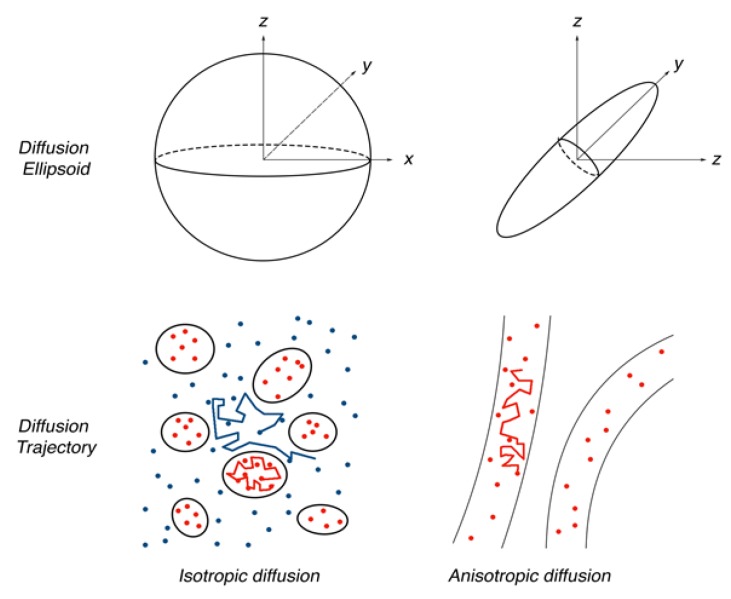
Example of ellipsoidal diffusion tensors depending on the behaviour of water molecules in different tissue microstructures.

**Figure 2 brainsci-08-00023-f002:**
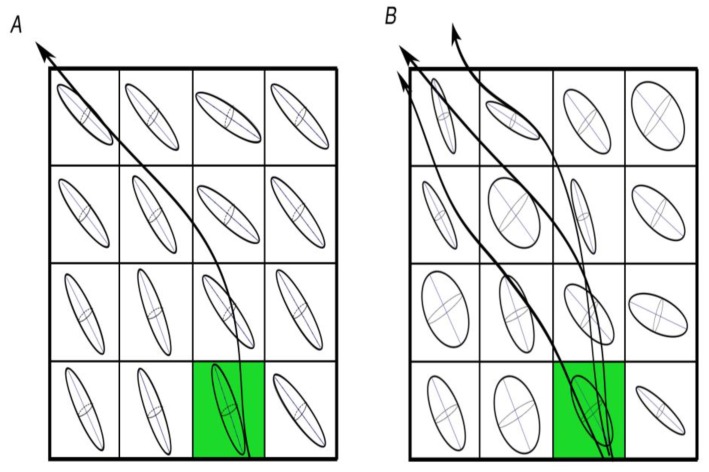
Visual representation of the differences between deterministic (**A**) and probabilistic; (**B**) approaches to tractography. Probabilistic approaches are considerably more extensive and computationally demanding, as they track all orientations in all voxels adjacent to the seed point (green).

**Figure 3 brainsci-08-00023-f003:**
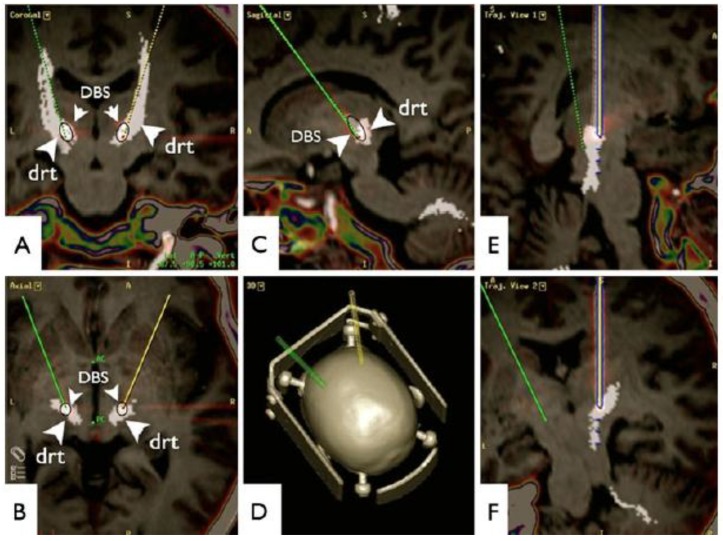
Diffusion tensor imaging fibre tracking-based targeting of the dentato-rubro-thalamic tract (DRT). Superimposition with a postoperative helical computed tomography (artefact from deep brain stimulation electrodes, DBS). (**A**) Coronal; (**B**) axial; (**C**) sagittal views; (**D**) 3D; (**E**,**F**) Trajectory views of left electrode path. Reprinted from reference [33] with permission of Springer.

**Figure 4 brainsci-08-00023-f004:**
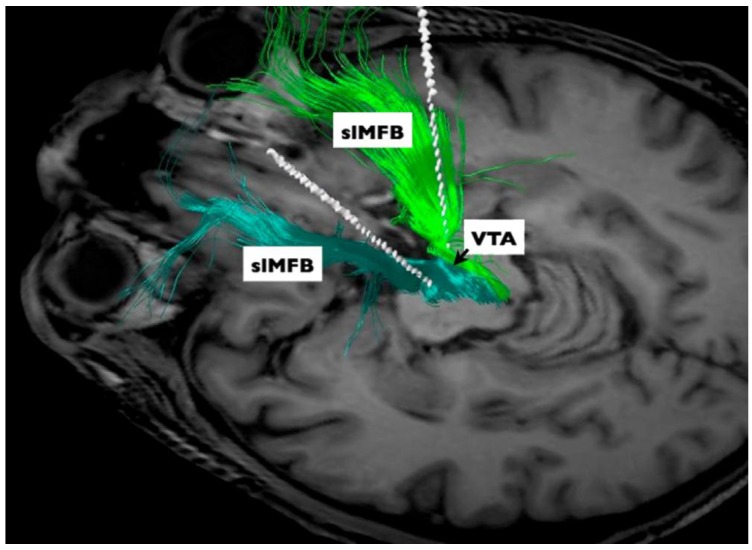
Medial forebrain bundle. Diffusion tensor imaging-based patient individual planning of bilateral supero-lateral medial forebrain bundle (slMFB) deep brain stimulation. Three-dimensional rendering as seen from posterior and superior left includes final DBS electrode positions (white rods). VTA, ventral tegmental area. Reprinted from Schlaepfer et al. [50]; with permission from Elsevier.

**Figure 5 brainsci-08-00023-f005:**
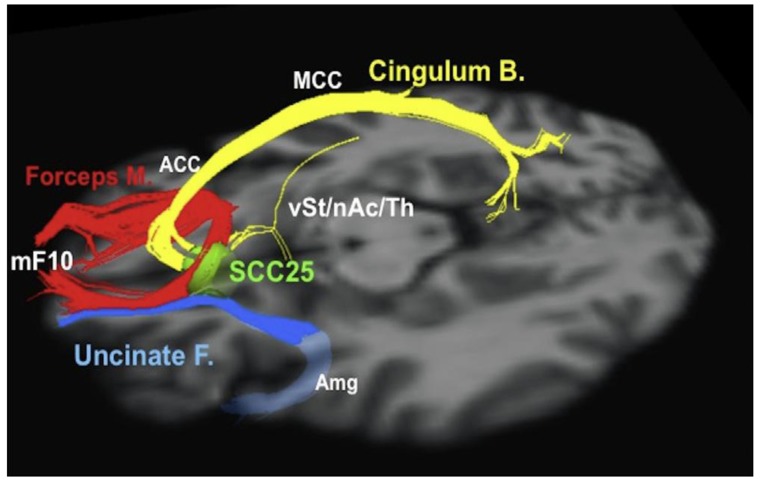
Subgenual cingulum. Optimal subcallosal cingulate deep brain stimulation fibre bundle target template. Red: forceps minor. Blue: uncinate fasciculus. Yellow: cingulate bundle. ACC, anterior cingulate cortex; Amg, amygdala; Cingulum B., cingulum bundle; Forceps M., forceps minor; MCC, middle cingulate cortex; mF10, medial frontal (Brodmann area 10); nAc, nucleus accumbens; SCC25, subcallosal cingulate cortex (Brodmann area 25); Th, thalamus; Uncinate F., uncinate fasciculus; vSt, ventral striatum. Reprinted from Riva-Posse et al. [57] with permission from Elsevier.

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
