# Peer review of "The Emerging Role of Tractography in Deep Brain Stimulation: Basic Principles and Current Applications"

_brainsci, 2018, doi:10.3390/brainsci8020023_

Round 1

Reviewer 1 Report

Figure 2, need to indicate what B corresponds to.

- Although Tractography is widely used to describe the potential location of white matter tracts, it is based on several assumptions that pose limitations to the method. Limitations on the translational use of tractography could have been included.

- Recent literature has described the potential use of tractography for targeting and DBS programming on Parkinson's Disease. Authors did not include recent relevant literature on this topic.

Author Response

Figure 2, need to indicate what B corresponds to.

This has now been indicated in the legend.

- Although Tractography is widely used to describe the potential location of white matter tracts, it is based on several assumptions that pose limitations to the method. Limitations on the translational use of tractography could have been included.

Some of the limitations of tractography have now been included (lines 103-110).

- Recent literature has described the potential use of tractography for targeting and DBS programming on Parkinson's Disease.. Authors did not include recent relevant literature on this topic.

In the initial version, our goal was to use tremor and depression as practical examples of applications in which tractography has been used to refine DBS targeting. We have now included some studies on PD, as suggested by the reviewer (lines 22-247). 

Reviewer 2 Report

This is a very interesting and important review that is concerned with the used of Diffusion Tensor Magnetic Resonance Imaging Tractography (DTI FT)  to assist deep brain stimulation targeting. The focus of this rather brief report is the use of DTI FT in tremor (targeting the DRT) and depression. The authors mention some of the important work in the field and stay selective in other parts. While I am in favor of a publication, I think it needs revisions in some major points: 

Concerns:

 - there are other than deterministic approaches that guide DBS. It needs to be mentioned that the FDA/CE - approved planing platforms exquisitely use deterministic approaches. The work |27-28] use probabilistic approaches (as an add on to compare to deterministic approach later on). They should discuss the principles of probabilistic and global racking algorithms in brief, since the deterministic approach has been accused of being anatomically inacurate. 

 - there needs to be a discussion on the anatomical truth of DTI FT. It is a mathematical process (FACT, HAFT etc.) that is misused to show anatomy. It can have has serious flaws with regards to kissing, crossing branching fibers that should be addressed. 

 - Likewise there should be some discussion about the spatial accuracy. DTI has a reduced resolution depending on the scanning and sequencing of the post-processing. - Geometrical inaccuracies need to be addressed especially since the DT sequence will be heavily distorted. This is of special importance in the cg25 region close to the cc with some centimeters of distortion possible. 

- The work on STN and the hyperdirect pathway should at least be mentioned e.g. Petersen MV, Lund TE, Sunde N, Frandsen J, Rosendal F, Juul N, et al. Probabilistic versus deterministic tractography for delineation of the cortico-subthalamic hyperdirect pathway in patients with Parkinson disease selected for deep brain stimulation. J Neurosurg. 2016 Jul 8;:1–12. 

 - The slMFB (the target region is slMFB and not MFB) has been introduced as a possible target for major depression [29, published 2013 in a small case series targeted with DT FT)] BEFORE DTI FT - targeting [27-28] was used for the four fiber tracts necessary to get efficacious cg25 targeting [26]. First publication for cg25 in 2014 (retrospective series [27]), prospectively used and published in 2017 [28]. Thus mentioning "A second set of studies using tractography based DBS [...]" is somewhat anachronistic and should be revised. The slMFB actually is the first ever introduced tractographically designed target region. This target was published (2011) showing in a simulation that other target regions for TRD would likewise influence the distal slMFB system. The rationale for this bundle has thus been described. 

The missing reference underpinning this is: Coenen VA, Schlaepfer TE, Maedler B, Panksepp J. Cross-species affective functions of the medial forebrain bundle-implications for the treatment of affective pain and depression in humans. Neuroscience and Biobehavioral Reviews. 2011 Oct;35(9):1971–81. 

The anatomical substrate of slMFB DBS actually is NOT unclear. The rational has been thoroughly described and the same group has in detail described the MFB and ATR as a reproducible fiber system that has been adapted by other authors [36] as well. 

Missing reference: Coenen VA, Panksepp J, Hurwitz TA, Urbach H, Mädler B. Human Medial Forebrain Bundle (MFB) and Anterior Thalamic Radiation (ATR): Imaging of Two Major Subcortical Pathways and the Dynamic Balance of Opposite Affects in Understanding Depression. J Neuropsychiatry Clin Neurosci. American Psychiatric AssociationArlington, VA; 2012;24(2):223–36. 

Minor points: 

 page 4: The DRT "mainly" origins in the dentate nucleus (there are parts from the emboliform and globose nuclei) . 

Figure 3: DTI fiber tracking-based "targeting" of the dentato-rubro-thalamic [...].

Author Response

Concerns:

 - there are other than deterministic approaches that guide DBS. It needs to be mentioned that the FDA/CE - approved planing platforms exquisitely use deterministic approaches. The work |27-28] use probabilistic approaches (as an add on to compare to deterministic approach later on). They should discuss the principles of probabilistic and global racking algorithms in brief, since the deterministic approach has been accused of being anatomically inacurate. 

These aspects have now been addressed (lines 110-119; 160-183).

 - there needs to be a discussion on the anatomical truth of DTI FT. It is a mathematical process (FACT, HAFT etc.) that is misused to show anatomy. It can have has serious flaws with regards to kissing, crossing branching fibers that should be addressed. 

These aspects are now discussed in the tractography section of our article.

 - Likewise there should be some discussion about the spatial accuracy. DTI has a reduced resolution depending on the scanning and sequencing of the post-processing. - Geometrical inaccuracies need to be addressed especially since the DT sequence will be heavily distorted. This is of special importance in the cg25 region close to the cc with some centimeters of distortion possible. 

This has bee addressed in lines 155-157

- The work on STN and the hyperdirect pathway should at least be mentioned e.g. Petersen MV, Lund TE, Sunde N, Frandsen J, Rosendal F, Juul N, et al. Probabilistic versus deterministic tractography for delineation of the cortico-subthalamic hyperdirect pathway in patients with Parkinson disease selected for deep brain stimulation. J Neurosurg. 2016 Jul 8;:1–12. 

A few paragraphs on the use of tractography in PD, including the study mentioned above, has been incorporated to the text (lines 222-247).

 - The slMFB (the target region is slMFB and not MFB) has been introduced as a possible target for major depression [29, published 2013 in a small case series targeted with DT FT)] BEFORE DTI FT - targeting [27-28] was used for the four fiber tracts necessary to get efficacious cg25 targeting [26]. First publication for cg25 in 2014 (retrospective series [27]), prospectively used and published in 2017 [28]. Thus mentioning "A second set of studies using tractography based DBS [...]" is somewhat anachronistic and should be revised. The slMFB actually is the first ever introduced tractographically designed target region. This target was published (2011) showing in a simulation that other target regions for TRD would likewise influence the distal slMFB system. The rationale for this bundle has thus been described. 

The missing reference underpinning this is: Coenen VA, Schlaepfer TE, Maedler B, Panksepp J. Cross-species affective functions of the medial forebrain bundle-implications for the treatment of affective pain and depression in humans. Neuroscience and Biobehavioral Reviews. 2011 Oct;35(9):1971–81. 

The anatomical substrate of slMFB DBS actually is NOT unclear. The rational has been thoroughly described and the same group has in detail described the MFB and ATR as a reproducible fiber system that has been adapted by other authors [36] as well. 

Missing reference: Coenen VA, Panksepp J, Hurwitz TA, Urbach H, Mädler B. Human Medial Forebrain Bundle (MFB) and Anterior Thalamic Radiation (ATR): Imaging of Two Major Subcortical Pathways and the Dynamic Balance of Opposite Affects in Understanding Depression. J Neuropsychiatry Clin Neurosci. American Psychiatric AssociationArlington, VA; 2012;24(2):223–36. 

These considerations have been addressed in lines 260-277. Missing references were added to the text.

In our view, more important than the chronology, the slMFB was a DTI-generated target whereas DTI has been used to refine a pre-existing target.  

The term “second set” of studies was not written in a chronological context. The meaning was “alternative set”. The term has now been removed form the text.

Minor points: 

 page 4: The DRT "mainly" origins in the dentate nucleus (there are parts from the emboliform and globose nuclei) . 

Figure 3: DTI fiber tracking-based "targeting" of the dentato-rubro-thalamic [...].

These have now been corrected. 

Reviewer 3 Report

Line 90: typo in the word tractography

Line 198: "experiences" should be "experienced"

Overall, this is a very nice review explaining tractography and its emerging role in DBS procedures. Use of figures to describe concepts was very helpful. Caveats and limitations were clearly and appropriately addressed. 

Author Response

Line 90: typo in the word tractography

Line 198: "experiences" should be "experienced"

Overall, this is a very nice review explaining tractography and its emerging role in DBS procedures. Use of figures to describe concepts was very helpful. Caveats and limitations were clearly and appropriately addressed. 

We appreciate the comment. The pointed typos have been corrected.     

Round 2

Reviewer 2 Report

They have addressed my concerns. Thank you! Nice paper!